# Patient use of pulse oximetry to support management of COVID-19 in Greater Manchester: A non-randomised evaluation using a target trial approach

Fernando Rubinstein[1], Richard Williams[2], Jo Dumville[3], Binita Kane[4], William Whittaker[5], Peter Bower[6], Evangelos Kontopantelis[2]*

1 The Global Health Network, Centre for Tropical Medicine and Global Health, University of Oxford, Oxford, United Kingdom, 2 NIHR Applied Research Collaboration Greater Manchester, Division of Informatics, Imaging and Data Science, Faculty of Biology, Medicine and Health, Faculty of Biology, Medicine and Health, University of Manchester, Manchester Academic Health Science Centre, Manchester, United Kingdom, 3 NIHR Applied Research Collaboration Greater Manchester, Division of Nursing, Midwifery and Social Work, Faculty of Biology, Medicine and Health, Faculty of Biology, Medicine and Health, University of Manchester, Manchester Academic Health Science Centre, Manchester, United Kingdom, 4 Manchester University Foundation Trust, Manchester, United Kingdom, 5 NIHR Applied Research Collaboration Greater Manchester, Manchester Centre for Health Economics, Division of Population Health, Health Services Research and Primary Care, Faculty of Biology, Medicine and Health, Faculty of Biology, Medicine and Health, University of Manchester, Manchester Academic Health Science Centre, Manchester, United Kingdom, 6 NIHR Applied Research Collaboration Greater Manchester, Division of Population Health, Health Services Research and Primary Care, Faculty of Biology, Medicine and Health, Faculty of Biology, Medicine and Health, University of Manchester, Manchester Academic Health Science Centre, Manchester, United Kingdom

* e.kontopantelis@manchester.ac.uk

**Data Availability Statement:** GMCR data has been made available in de-identified format for research but only in a secure environment. Research protocols (and other data requests) must be

## Abstract

### Introduction

The pandemic saw widespread use of home pulse oximeters to patients diagnosed with COVID-19 to support early detection of low oxygen saturation levels and appropriate care. Rapid implementation made conventional evaluation challenging, highlighting the need for rigorous non-randomised methods to support decision-making about future use of these technologies. We used routine data to explore the benefits of pulse oximetry in Greater Manchester, under the 'COVID-19 oximetry at home' (CO@h) programme.

### Methods

We used data from the Greater Manchester Secure Data Environment and defined study parameters using a 'target trial' model to compare patients receiving pulse oximetry under the CO@h programme, with matched controls using various comparator groups. Primary outcomes were unplanned hospitalisation and all-cause mortality. This study is based on data from the Greater Manchester Care Record (GMCR), using anonymised, routinely collected data provided in a de-identified format for research. Informed written consent is needed for primary care patient data to be collected for service improvement and research, before data extraction to the GMCR. The study was approved under protocol GMCR RQ-

approved by GMCR and we are unable to make data available outside that process. For more information on access to this data resource please contact gm.sde@healthinnovationmanchester.com.

**Funding:** This research was funded by the National Institute for Health and Care Research (NIHR) Applied Research Collaboration Greater Manchester (ARC-GM). The views expressed in this publication are those of the authors and not necessarily those of the National Institute for Health and Care Research or the Department of Health and Social Care.

**Competing interests:** RW, JD, PB, FR and EK were supported by the NIHR Applied Research Collaboration Greater Manchester (NIHR200174) and RW and JD by the NIHR Manchester Biomedical Research Centre (NIHR203308). As explained in the data availability section, there are restrictions using patient level data and we cannot freely share the data used in this paper through the journal.

048, on 12/05/2022. As indicated by the University of Manchester ethics decision tool, formal ethical approval was not required for this study.

## Results

The adjusted odds ratios for an unplanned hospitalisation were higher among patients receiving pulse oximetry: OR 1.86 (95% CI 1.54–2.25) at 28 days, 1.5 (95% CI 1.3–1.74) at 90 days and 1.63 (95% CI 1.44–1.83) at 1 year. Overall odds of mortality were lower among patients receiving pulse oximetry: adjusted ORs of 0.5 (95% CI 0.25–0.98) at 28 days, 0.5 (95% CI 0.32–0.78) at 90 days and 0.58 (95% CI 0.44–0.76) at 1 year. The results were robust to different comparison groups.

## Conclusion

Use of pulse oximetry at home under the CO@h programme, through the resulting prioritisation for appropriate care, was associated with a higher frequency of unplanned admissions and a reduction in the risk of mortality up to 1 year later. Therefore, it is likely effective for early detection of clinical deterioration and timely intervention among patients with COVID-19. Further research is needed to understand whether this is a cost-effective use of healthcare resources.

## Introduction

The national 'COVID-19 oximetry at home' (CO@h) programme was launched by NHS England in November 2020, providing pulse oximeters to patients diagnosed with COVID-19 and at risk of deterioration due to silent hypoxia. The use of pulse oximeters at home allowed for early detection of low oxygen saturation levels and prompted escalation to the correct care setting. Responses included same-day clinician review at home or if more urgent, conveyance by ambulance to hospital. If patients did not show signs of clinical deterioration within 14 days of onset of symptoms, they were discharged with safety-netting advice.

The rapid impact of COVID-19 on both clinical services and research delivery made conventional randomised evaluation of innovations such as CO@h challenging. Some evaluations have used non-randomised comparative designs to assess effectiveness. Assessments at the population level often provided limited evidence, due to low uptake, variability of the CO@h services offered, and barriers to data access [1]. Other non-randomised studies used patient-level data, comparing patients receiving CO@h with controls. Patients receiving CO@h following accident and emergency (A&E) attendance had odds of subsequent A&E attendance and emergency admission 37% and 59% higher than controls, and odds of 28-day mortality 52% lower (n = 15,621, of which 639 were the CO@h group) [2]. Another patient-level study in a community led service compared outcomes between COVID-19 patients admitted to 2 hospitals, in a group receiving CO@h later escalated to hospital, and those directly admitted to hospitals without CO@h (n = 748, of which 115 were the CO@h group). Results showed that receipt of CO@h was associated with a 79% reduction in the 30-day odds for hospital mortality (adjusted OR 0.21, 95% CI 0.08 to 0.47), as well as a shorter hospital stay for those admitted, and a slightly lower risk for admission to intensive care [3]. These studies provide preliminary evidence on the effectiveness of CO@h and suggest that appropriate escalation to hospital triggered by monitoring resulted in better patient outcomes. An interim analysis of an ongoing

pragmatic trial assessing the addition of oximetry to an existing service (COVID Watch, which involved COVID-19 patients receiving text messages about dyspnea and nurse support) showed at 30 days that there were no between-group differences in the number of days they were alive and out of the hospital, hospitalisations or deaths. However, the number of events in this interim analysis was low [4].

The current evidence on the benefits of oximetry is far from definitive, which is problematic in the context of evolving pandemic threats. We took a target trial approach to evaluate the comparative effectiveness of CO@h on hospital admissions and mortality. We add to the evidence base using routine data on the implementation of CO@h in Greater Manchester, with a large sample of patients and longer-term follow up than previous assessments.

## Materials and methods

### Study design

Greater Manchester in the Northwest of England has an ethnically diverse population with high levels of deprivation and consists of ten localities with a total population of 2.7 million. CO@h was implemented widely for monitoring clinical deterioration in patients diagnosed with COVID-19 in Greater Manchester. Referrals were taken from all healthcare settings including primary care, urgent care, online systems ambulance services, nursing homes and hospital discharges. However, use of CO@h was not mandatory, and it was expected that its adoption would show substantial variability, allowing identification of potentially eligible patients who did and did not receive the intervention. The model used was the OxyWatch portable non-invasive oximeter, by ChoiceMMed (https://www.win-health.com/oxywatch-fingertip-pulse-oximeter). A bespoke dashboard was built in the Greater Manchester Care Record (GMCR—a joint electronic patient record across the city region). However, this was not used in all ten localities as some areas opted to use existing systems to record activity.

We used data from the Greater Manchester Secure Data Environment (GM SDE) which includes linked primary care and specialist health records [5]. Data were obtained on 6/4/2023 and were accessed from that date until 26/1/2024. We undertook a 'target trial' emulation approach, which applies design principles of trials to the analysis of non-randomised comparative data, explicitly matching the analysis to the trial it is emulating [6–8]. Design features of the target trial and the emulated target trial are specified in Fig 1 and we report our study in line with published guidelines [9].

When first set up in September 2020, the original criteria for eligibility for CO@h were:

a. patients with COVID-19 who were symptomatic and aged 65+(later extended to 50+) or

b. 'clinically extremely vulnerable' to COVID-19 or

c. identified by a clinician as at high risk of deterioration from acute COVID-19.

To emulate participant eligibility, we identified individuals in the GM SDE between 50–85 years of age with a positive COVID-19 test (February 2020 to June 2022) who received CO@h and were not admitted to the hospital or who died within 24 hours of diagnosis.

### Treatment assignment and follow-up

Instead of random assignment, we initially retrieved data from anonymised patients identified in the GM SDE as enrolled in CO@h. These were to be matched to similar patients in the GM SDE who were not identified as enrolled in CO@h.

The identification of suitable comparator patients was complicated by the fact that only 2 of the 10 areas in Greater Manchester routinely identified those enrolled in CO@h in the GM

| Target trial specification | Target trial emulation |
|---|---|
| **Eligibility criteria** ||
| People from community settings:<br>• clinical diagnosis and positive COVID-19 test<br>• between 50–85 years of age<br>• not admitted or died within 24 hours | Same as the target trial |
| **Treatment strategies** ||
| *Intervention*. 'CO@h' programme<br>*Comparator*. Not recorded as 'CO@h' | Same as the target trial |
| **Treatment assignment** ||
| Use of any appropriate randomisation mechanism | Matching groups according to age, gender, date, number of days since testing positive, ethnicity, comorbidity and deprivation, admissions in year prior to COVID-19 diagnosis |
| **Outcomes** ||
| Primary outcomes<br>• any unplanned hospitalisation (28, 90 days and 1 year) after COVID-19 diagnosis<br>• all-cause mortality<br><br>Secondary outcomes:<br>• Hospital stay for those admitted | Same as the target trial |
| **Follow up** ||
| Up to 12 months since COVID-19 diagnosis | Same as the target trial |
| **Causal contrasts** ||
| Intention-to-treat analysis | By matching, we sought baseline balance and performed an analogue of intention-to-treat |
| **Statistical analysis** ||
| Direct comparison of groups by assigned treatments | Same as the target trial<br><br>Use of multivariable regression to estimate unadjusted and adjusted measures of effect<br><br>Causal inference and average treatment effects based on absolute risk differences |

**Fig 1. Specification and emulation of a target trial.**

SDE. Therefore, we could be sure that comparator patients in those 2 areas would not have received CO@h. However, in the remaining 8 areas, people in those areas not coded as enrolled in CO@h would have included some people who had been enrolled on CO@h and others who were not. Therefore, the comparator group would have a degree of 'contamination'.

We therefore undertook 3 analyses:

a. Comparison A compared those identified as enrolled in CO@h in 2 areas, with similar patients across all 10 areas

b. Comparison B compared those identified as enrolled in CO@h in 2 areas, with similar patients across the remaining 8 areas

c. Comparison C compared those identified as enrolled in CO@h in 2 areas, with similar patients from those 2 areas only

Comparison A and B are both influenced by contamination of the comparator group, although rates of contamination would be higher in B than in A. Comparison C is not affected by contamination, but may be more vulnerable to other selection biases in those areas which determined who was and who was not enrolled in CO@h.

Control patients were exactly matched by gender, age (within 1 year) and date of positive COVID-19 test (within 30 days) to participants who were not enrolled (ratio of 15 comparators per intervention patient). To maximise comparability on additional characteristics, we then used weighted coarse exact matching [10] on the categories of the following variables: Charlson score quartiles, ethnicity category, deprivation quintiles (as measured by the Index of Multiple Deprivation), and admissions within the 12 months prior to COVID-19 diagnosis. After defining the treatment variable, coarse exact matching involves three steps: (1) grouping each variable in defined categories (2) sorting all units into strata, each of which has the same values of the coarsened variable (3) removing units in any strata that do not include at least one treated and one comparator. The result is typically less model-dependent, has lower bias and increased efficiency compared with other matching methods, and can be combined with model based strategies (such as inverse probability of treatment weighting) to estimate a causal effect [10, 11]. Conditional on this matching, it is expected that the distribution of observed baseline characteristics would be similar between groups. Balance was formally evaluated by estimating standardised mean differences and variance ratios.

In a target trial, the natural baseline would be the time when the intervention is randomly allocated. The optimal way to emulate the target trial was to define time zero for the intervention as the date when a patient received the intervention as identified in the GM SDE, and the same start date for matched controls. Each patient was then 'followed up' to 1 year after the index date or until June 2022 if this period was shorter. The follow-up ended at that time or until death if the patient died within the 12 months after COVID-19 diagnosis.

## Outcomes

Primary outcomes were (a) unplanned hospitalisation for any reason and (b) all-cause mortality. Both were reported at 28 days, 90 days and 1 year after COVID-19 diagnosis. A secondary outcome was length of hospital stay for those admitted.

## Statistical analysis

We describe participant characteristics in the intervention and comparator groups using measures of central tendency and dispersion for continuous variables and proportions for

categorical variables. We compared the distribution of all covariates between the groups using t-tests or non-parametric tests for continuous variables and chi$^2$ test for proportions.

We assessed our primary outcomes using multiple logistic regression and survival models (competing risk models for unplanned hospitalisations and Cox regression for all-cause mortality). We estimated odds ratios and hazard/subhazard ratios of the intervention on the primary outcomes, adjusting for age, gender, ethnicity, Charlson quartiles, IMD quintiles and admissions within the 12 months prior to COVID-19 diagnosis, as there are likely to be varying risks for unplanned admissions and mortality for these groups that may confound the relationship between CO@h and outcomes of interest. Finally, we used augmented inverse probability of treatment weighting [12, 13] to estimate the average treatment effects of CO@h on mortality. This estimator requires (1) specification of a binary regression model for the propensity score, and (2) specification of a regression model for the outcome variable. This so-called 'double robustness' allows the estimator to remain consistent for the average treatment effect even if either the propensity score model or the outcome regression is mis-specified but the other is properly specified. It has been shown that the augmented inverse probability weighted estimator is superior to other methods of model-based matching [12]. Length of stay was analysed using Poisson or negative binomial regression, if overdispersion was evident visually and statistically (the variance was greater than the mean). Data was complete for all variables except for ethnicity, for which a "missing" category was used. All analyses were done with Stata v18.

## Results

Fig 2 shows the grouping of patients into the intervention and comparator arms following identification in the GM SDE and application of the target trial criteria. Table 1 shows the characteristics of the matched samples. Formal testing of comparability resulted in standardised mean differences below 0.1 and all variance ratios ~ = 1.

The primary outcome analysis are summarised in Table 2 (admissions) and Table 3 (mortality). The adjusted odds ratios for an unplanned admission following COVID-19 diagnosis were between 50% and 80% higher among CO@h patients: OR 1.86 (95% CI 1.54–2.25) at 28 days, 1.5 (95% CI 1.3–1.74) at 90 days and 1.63 (95% CI 1.44–1.83) at 1 year in comparison A (Fig 3). The overall strength and direction of effect was consistent over the 3 comparisons, although the comparison at greatest risk of selection bias (comparison C) shows the smallest impacts on admissions.

Overall odds of mortality were lower in CO@h patients, with adjusted ORs of 0.5 (95% CI 0.25–0.98) at 28 days, 0.5 (95% CI 0.32–0.78) at 90 days and 0.58 (95% CI 0.44–0.76) at 1 year in comparison A (Fig 4). Consistent results were seen in survival analysis, with a median follow up of 309 and 291 days for intervention and control respectively. Again, the overall strength and direction of effect was consistent over the 3 comparisons, although the comparison at greatest risk of selection bias (comparison C) showed the smallest impacts.

The estimated adjusted average treatment effect of CO@h on the risk of death at 28, 90 days and one year was between -0.2% and -1%, an expected reduction of between 2 deaths per 1000 patients at 28 days, 6 at 90 days and 10 at 1 year. In secondary outcomes, no difference was found in the length of stay of those admitted within 90 days of COVID-19 diagnosis between the groups (overall mean length of stay 9 days, difference -1 (95% CI -2.7 to 0.59).

## Discussion

### Statement of principal findings

Using target trial emulation with routine NHS data, our analyses suggested that CO@h was associated with a higher frequency of unplanned admissions and a reduction in mortality up

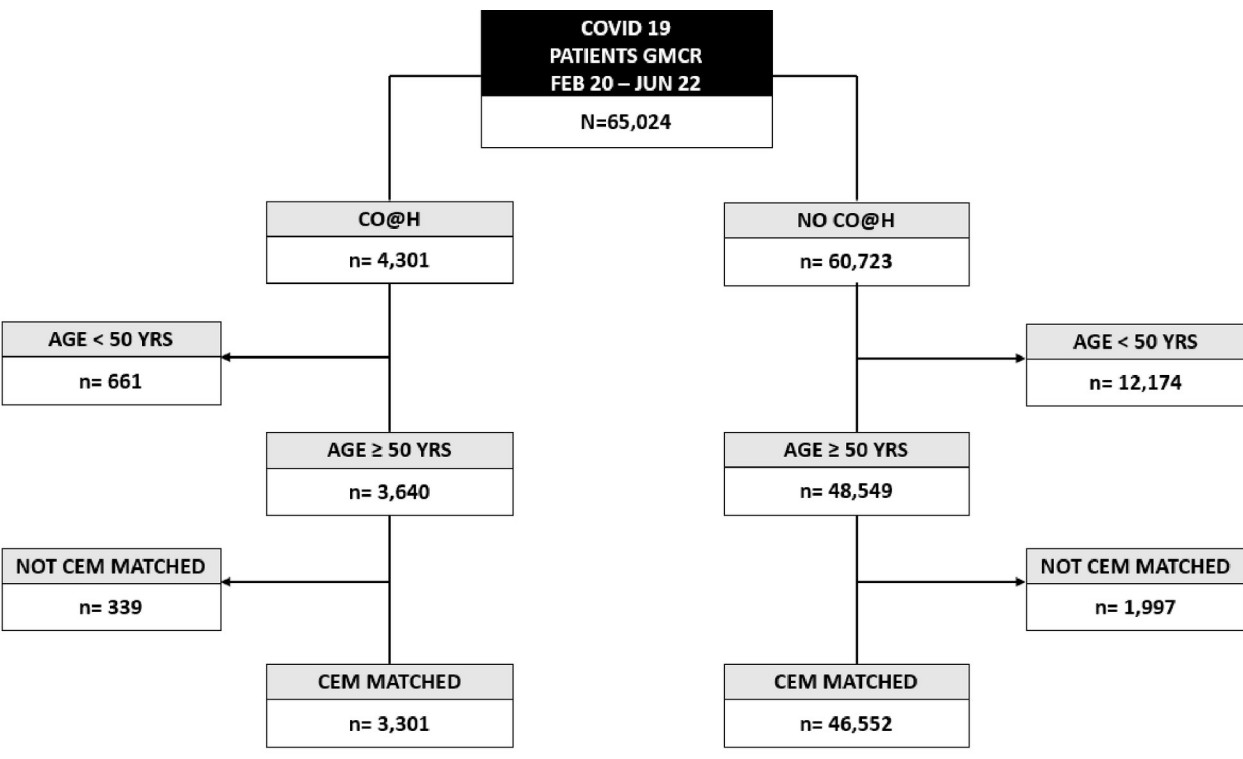

CEM: Coarse exact matching

**Fig 2. Flowchart of patient selection–comparison A.**

to 1 year later, after matching and adjustment for clinical and sociodemographic characteristics. In absolute terms, this difference translates to 2–10 deaths averted per 1000 patients.

## Strengths and limitations of the study

Using local routine data, we were able to identify a proportion of the patients in the region receiving the intervention, as well as data on diagnosis, admissions and mortality dates, demographic and clinical characteristics, local setting information and use of hospital services. We adopted a target trial approach to enhance rigour through pre-specification of our eligibility criteria and analysis. Target trial emulation not only includes methods to balance comparisons to limit bias and confounding but also considers important additional elements of the trial design, such as defining clear eligibility criteria of potential participants, treatment assignment and deployment, definition of 'time zero' (starting point of the intervention and control group) and follow up, and definition and measurement of prespecified outcomes [6, 7]. Coarse exact matching achieved a good balance between groups, and we controlled for numerous potential confounders. However, several limitations also exist. Although the methods we used are amongst the best available, we can never be certain on causality, and the quality of evidence can never match that of a randomised controlled trial. No matter how many variables are included in the modelling to account for measured confounding, there are always risks: not achieving good enough balance for the important drivers of the outcome of interest (since matching is a dimension reduction method); the possibility of residual confounding by disease severity or other factors (unmeasured confounding), for example, weight or BMI; finally, these are observational data from health care records, and their quality is, more often than not, not

**Table 1. Demographic and clinical characteristics—matched samples.**

| | Comparison A | | Comparison B | | Comparison C | |
| --- | --- | --- | --- | --- | --- | --- |
| | Comparator | CO@h | Comparator | CO@h | Comparator | CO@h |
| **n** | 46,552 | 3,301 | 37,306 | 3,063 | 8,788 | 3,018 |
| **Mean age (SD)** | 64.64 (9.82) | 64.08 (9.99) | 64.42 (9.8) | 64.11 (9.9) | 64.9 (9.8) | 64.1 (9.96) |
| **Age> 65** | 46.9% | 46.4% | 46.4% | 46.4% | 47% | 46.2% |
| **Male** | 49.1% | 49.3% | 49.8% | 49.6% | 49.3% | 49.6% |
| **White** | 83.4% | 83.9% | 84.6% | 84.5% | 85.6% | 85.5% |
| **Asian** | 11.6% | 11.1% | 10.9% | 10.4% | 10.8% | 10.6% |
| **Black** | 0.9% | 0.8% | 0.6% | 0.63% | 0.6% | 0.5% |
| **Mixed—Other** | 2.1% | 2.2% | 2.1% | 2.1% | 1.5% | 1.6% |
| **Not Known** | 2.0% | 2.1% | 1.8% | 1.9% | 1.5% | 1.7% |
| **CHD** | 9.9% | 9.2% | 10% | 9.1% | 8.6% | 8.9% |
| **COPD** | 6.2% | 7.5% | 6.1% | 7.3% | 6.2% | 7.3% |
| **Asthma** | 18.5% | 20.1% | 18.4% | 19.3% | 17.2% | 19.3% |
| **Diabetes** | 32.4% | 32.1% | 32.1% | 31.6% | 29.4% | 31.2% |
| **Charlson quartile 1** | 22.8% | 24.7% | 24.7% | 24.8% | 22.3% | 25% |
| **Charlson quartile 2** | 26.9% | 26.1% | 26.5% | 26.2% | 27.2% | 25.9% |
| **Charlson quartile 3** | 31.7% | 30.9% | 31.1% | 28.4% | 32.3% | 31.2% |
| **Charlson quartile 4** | 18.5% | 18.3% | 18.2% | 16.8% | 18.2% | 17.8% |
| **IMD2019 quintile 1** | 34.9% | 34.6% | 34.5% | 35.7% | 34.5% | 34.9% |
| **IMD2019 quintile 2** | 16.7% | 16.7% | 16.8% | 16.5% | 16.4% | 16.6% |
| **IMD2019 quintile 3** | 13.7% | 14.5% | 14.5% | 14.2% | 14.3% | 14.3% |
| **IMD2019 quintile 4** | 18.1% | 16.7% | 16.7% | 16.4% | 17.3% | 16.7% |
| **IMD2019 quintile 5** | 16.6% | 17.7% | 17.5% | 17.2% | 17.5% | 17.4% |
| **Prior admissions** | 11% | 10% | 9.1% | 9.2% | 10% | 9% |

Comparison A: Cases from 2 localities vs controls from all 10 localities. Comparison B: Cases from 2 localities vs controls from all other 8 localities. Comparison C: All cases and controls from 2 localities

very high. It also is possible that some patients who were admitted to hospital and required oxygen were prescribed Dexamethasone and/or were part of the RECOVERY trial in 2021 [14]. However, by definition, the patients admitted to the CO@H service were not on oxygen and were considered low risk, thus it can be assumed that the vast majority of patients did not have access to COVID19 specific treatments. However, as we don't have prescribing data, we cannot confirm this.

Patients and professionals were aware of the receipt of CO@h and this awareness may have affected their behaviour (e.g. patients and their families using CO@h may have been more conscious of certain signs and symptoms or followed recommendations more strictly). This is

**Table 2. Comparative effectiveness of CO@h–admissions.**

| | Comparison A | | | Comparison B | | | Comparison C | | |
| --- | --- | --- | --- | --- | --- | --- | --- | --- | --- |
| **ADMISSIONS** | 28 days | 90 days | 1 year | 28 days | 90 days | 1 year | 28 days | 90 days | 1 year |
| **Crude OR** | 2.1 (1.75–2.51) | 1.71 (1.52–1.96) | 1.78 (1.61–1.98) | 1.93 (1.64–2.28) | 1.87 (1.62–2.16) | 1.75 (1.58–1.94) | 1.22 (1.0–1.5) | 1.53 (1.32–1.78) | 1.2 (1.05–1.36) |
| **Adjusted OR** | 1.86 (1.54–2.25) | 1.5 (1.3–1.74) | 1.63 (1.44–1.83) | 1.86 (1.57–2.21) | 1.8 (1.55–2.1) | 1.8 (1.58–2.03) | 1.36 (1.12–1.64) | 1.32 (1.11–1.54) | 1.35 (1.18–1.55) |
| **Absolute risk** | 3.9% vs 1.9% | 7.2% vs 4.4% | 11.3% vs 6.7% | 5.2% vs 2.8% | 7.1% vs 3.9% | 11% vs 6.3% | 5.2% vs 4.3% | 7.7% vs 5.1% | 11.1% vs 9.5% |

**Table 3. Comparative effectiveness of CO@h—mortality.**

| | Comparison A | | | Comparison B | | | Comparison C | | |
|---|---|---|---|---|---|---|---|---|---|
| MORTALITY | 28 days | 90 days | 1 year | 28 days | 90 days | 1 year | 28 days | 90 days | 1 year |
| Crude OR | 0.56 (0.25–1.1) | 0.55 (0.33–0.85) | 0.63 (0.49–0.82) | 0.55 (0.29–1.03) | 0.58 (0.38–0.89) | 0.64 (0.49–0.84) | 0.57 (0.26–1.15) | 0.44 (0.27–0.69) | 0.47 (0.35–0.63) |
| Adjusted OR | 0.50 (0.25–0.98) | 0.5 (0.32–0.78) | 0.58 (0.44–0.76) | 0.59 (0.31–1.12) | 0.5 (0.32–0.77) | 0.56 (0.42–0.74) | 0.68 (0.34–1.38) | 0.51 (0.32–0.81) | 0.56 (0.41–0.75) |
| Absolute risk | 0.2% vs 0.4% | 0.6% vs 1.2% | 1.7% vs 2.7% | 0.29% vs 0.54% | 0.64% vs 1.17% | 1.73% vs 2.68% | 0.3% vs 0.5% | 0.69% vs 1.55% | 1.7% vs 3.6% |

similar to any pragmatic trial, which would likely be the optimal approach for the evaluation of an intervention like CO@h. Lack of independent outcome validation using routine data may risk misleading estimates due to differential outcome ascertainment, However, we do not expect our measures to be influenced because they are routinely recorded for all patients.

Table 4 presents the characteristics of our study alongside published evaluations. Our analysis had the largest sample size and was able to assess outcomes up to 1 year. The studies used different criteria for patient eligibility, especially on the age groups included and the settings. Our study identified patients recorded as having been referred to the CO@h, matched on several baseline demographic and clinical characteristics with those who were not coded as receiving the intervention, and included patients 50 years or over. Other studies included much younger patients, starting at 18 years, and included patients attending A&E or admitted to the

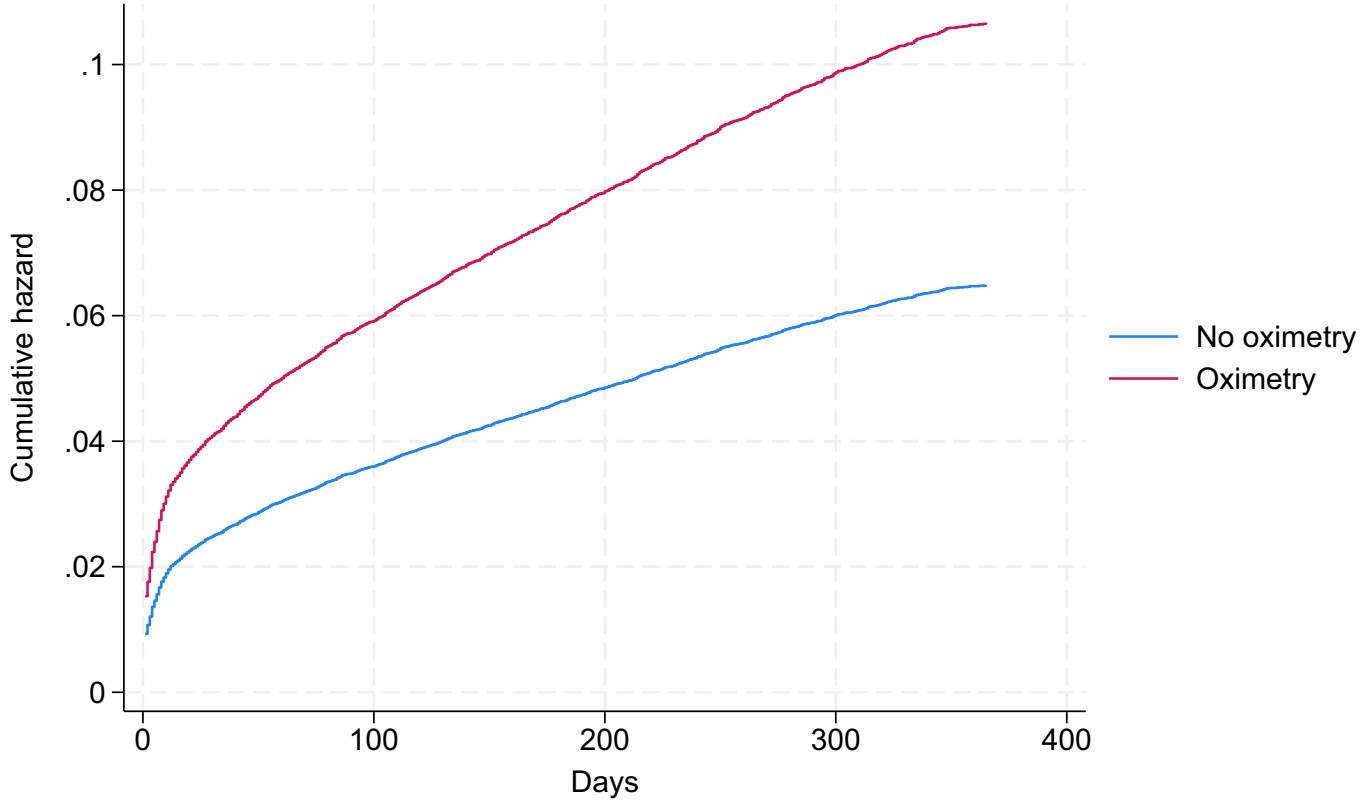

**Fig 3. Adjusted cumulative hazard for unplanned admissions at 1 year.**

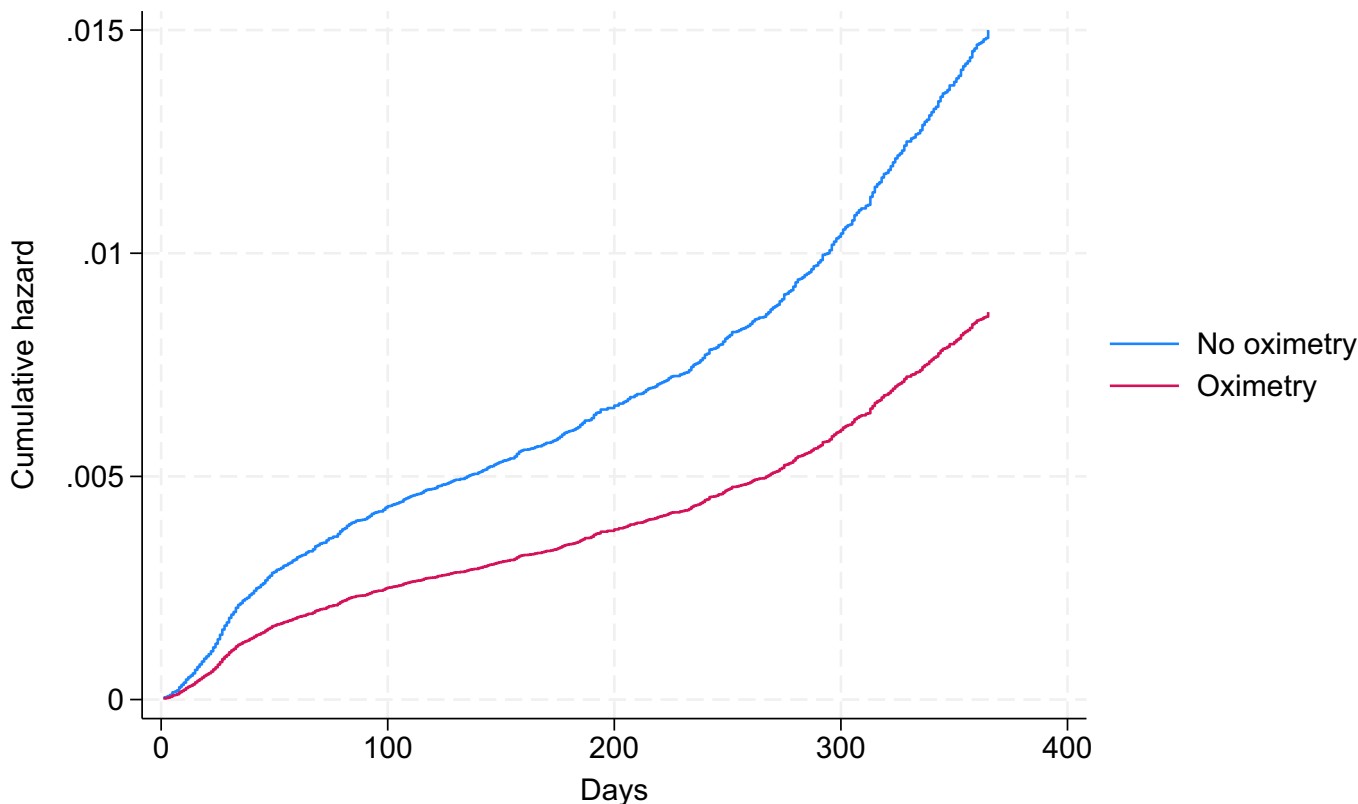

**Fig 4. Adjusted cumulative hazard for mortality at 1 year.**

**Table 4. Comparison of patient level studies evaluating CO@h.**

| | ICL | Boniface | Lee | Rubinstein |
|---|---|---|---|---|
| **Setting** | NHS services across England | 2 hospitals in North Hampshire | 6 hospitals in Pennsylvania, US | Greater Manchester |
| **Design** | Retrospective matched cohort | Retrospective cohort | Randomised trial | Retrospective matched cohort |
| **Patients** | n = 15,621 (18–80+) attending ED (639 oximetry 14,982 without) | n = 115 (20–85+) admitted after oximetry vs n = 633 admitted with no oximetry | >18, n = 611 Covid Watch plus oximetry vs n = 606 on Covid Watch alone | n = 49,853 (> = 50) (3301 oximetry, 46,552 without) |
| **Matching process** | Age, sex, ethnicity, terciles of IMD, BMI, ED index date, CEV status and days from COVID-19 test to ED index date | No matching | N/A | Matching on age, gender, ethnicity, diagnosis date, Charlson and IMD score, admissions in the previous year |
| **Stat methods / adjustment** | Weighted logistic regression | Logistic regression adjusting for age group, gender and comorbidities | Random assignment | Weighted logistic regression / survival analysis |
| **Mortality 28–30 days** | 1.5% vs 2.7%, OR 0.48 (0.25 to 0.93) | 5.2% vs 20.5% OR 0.21 (0.08 to 0.47) | 0.8% vs 0.5% p = NS | 0.2% vs 0.4% OR 0.5 (0.25–0.98) |
| **90 days** | | | | 0.6% vs 1.2% OR 0.5 (0.32–0.78) |
| **1 year** | | | | 1.7% vs 2.7% OR 0.58 (0.44–0.76) |
| **Unplanned admissions 28–30 days** | 22.7% vs 15.6% OR 1.59 (1.32–1.91) | 0% vs 8.7% | 7% vs 6.8% | 3.9% vs 1.9% OR 1.86 (1.54–2.25) |

hospital after having received (or not) CO@h prior to admission. These differences in patient characteristics and settings may account for differences in study results, suggesting that patients had dissimilar baseline risks if they attended the emergency department or were admitted to the hospital compared to patients enrolled in an ongoing successful monitoring programme or identified from a dataset of care records.

As well as the issues identified in coding of CO@h, patients tagged as having received the intervention were considered the intervention group, which means that those who were identified as eligible but who refused might have been included in the comparator group, therefore limiting the emulation of intention to treat analysis by not including them in the treated group as originally intended. This pragmatic assessment explored the impact of the policy of providing CO@h, and we had no data on patient adherence to the protocol. The coding issues identified highlight that importance of accurate and consistent coding of innovative interventions if the potential of routine NHS data for analysis of interventions (as opposed to epidemiological uses) is to be realised.

## Meaning of the study: Possible mechanisms and implications for clinicians or policymakers

Beaney et al. state that monitoring programs like CO@h should be viewed 'as a pathway to support appropriate escalation and decision-making for assessment or in-patient admission' [2]. Most studies have treated post-COVID-19 admissions as a 'process' outcome and mortality as an end point separately, although in the discussion it is usually mentioned that better use of services implies an expected greater number of admissions among patients on CO@h, and that may be part of the benefit on mortality due to timely detection of low oxygen saturation and escalation [15]. There is some evidence from this study that supports this view. In our study, the risk of post-COVID-19 admissions was higher for CO@h patients, considering death as a competing event. Although timely in-patient hospital admissions might be a partial mediator of the protective effect of CO@h on mortality, a post hoc sub-group exploration found a greater reduction in adjusted mortality risk among those patients who did *not* have an in-patient admission (adjusted OR 0.27 (95% CI 0.13–0.58) and more uncertainty in mortality risk among those who had at least one unplanned in-patient admission post-COVID-19 (adjusted OR 0.68 (95% CI 0.39–1.18). These findings may suggest that the effect of the CO@h programme is more evident among patients presenting with a less severe clinical profile and not always mediated by timely admissions.

## Unanswered questions and future research

Our study adds to the developing evidence base on the benefits of CO@h, including a longer term assessment of outcomes. Key questions about this technology remain, including: a comprehensive assessment of cost-effectiveness (which would in turn require assessment of quality of life in patients which is not routinely collected); wider patient and carer impacts (including quality of life, treatment burden and satisfaction); impacts on the workforce; and wider impacts on hospital capacity and efficiency. Further analysis on subgroups may be indicated, and would be important in the context of wider evidence showing differential accuracy in the technology associated with levels of skin pigmentation [16]. Work could also usefully explore the involvement of other services in supporting people on CO@h that might be supporting reduced risks to health without admission, as well as more general barriers and facilitators to the implementation of CO@h which might be relevant to other innovations based on remote monitoring such as virtual wards [17–19].

## Conclusion

CO@h, through accurate monitoring and the resulting appropriate care when needed, was associated with a higher frequency of unplanned admissions and a reduction in the risk of mortality, up to 1 year later. CO@h is likely an effective system for early detection of clinical deterioration and timely intervention among patients with COVID-19, although further research needed to understand whether this is a cost-effective use of healthcare resources.

## Supporting information

**S1 Checklist. PLOSOne human subjects research checklist.**
(PDF)

**S2 Checklist. STROBE checklist v4 cohort.**
(DOCX)

## Acknowledgments

This work uses data provided by patients and collected by the NHS as part of their care and support. The authors recognise the Greater Manchester Care Record (a partnership of Greater Manchester Health and Social Care Partnership, Health Innovation Manchester and Graphnet Health, on behalf of Greater Manchester localities) in the provision of data required to undertake this work.

## Author Contributions

**Conceptualization:** Fernando Rubinstein, Jo Dumville, Binita Kane, William Whittaker, Peter Bower, Evangelos Kontopantelis.

**Data curation:** Fernando Rubinstein, Richard Williams, Evangelos Kontopantelis.

**Formal analysis:** Fernando Rubinstein.

**Funding acquisition:** Jo Dumville, Evangelos Kontopantelis.

**Investigation:** Fernando Rubinstein, Richard Williams, William Whittaker, Peter Bower, Evangelos Kontopantelis.

**Methodology:** Fernando Rubinstein, Richard Williams, William Whittaker, Peter Bower, Evangelos Kontopantelis.

**Project administration:** Jo Dumville, William Whittaker, Peter Bower, Evangelos Kontopantelis.

**Resources:** Jo Dumville, Peter Bower, Evangelos Kontopantelis.

**Software:** Evangelos Kontopantelis.

**Supervision:** Jo Dumville, William Whittaker, Peter Bower, Evangelos Kontopantelis.

**Visualization:** Fernando Rubinstein.

**Writing – original draft:** Fernando Rubinstein.

**Writing – review & editing:** Richard Williams, Jo Dumville, Binita Kane, William Whittaker, Peter Bower, Evangelos Kontopantelis.

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
