## [Decision Letter · Decision Letter 0]

26 Aug 2024

PONE-D-24-30539Patient use of pulse oximetry to support management of COVID-19 in Greater Manchester: a non-randomised evaluation using a target trial approachPLOS ONE

Dear Dr. Kontopantelis,

Thank you for submitting your manuscript to PLOS ONE. After careful consideration, we feel that it has merit but does not fully meet PLOS ONE’s publication criteria as it currently stands. Therefore, we invite you to submit a revised version of the manuscript that addresses the points raised during the review process.

We look forward to receiving your revised manuscript.

Kind regards,

Felix Bongomin, MB ChB, MSc, MMed, FECMM

Academic Editor

PLOS ONE

Journal Requirements:

When submitting your revision, we need you to address these additional requirements. 1. Please ensure that your manuscript meets PLOS ONE's style requirements, including those for file naming. The PLOS ONE style templates can be found at https://journals.plos.org/plosone/s/file?id=wjVg/PLOSOne_formatting_sample_main_body.pdf and https://journals.plos.org/plosone/s/file?id=ba62/PLOSOne_formatting_sample_title_authors_affiliations.pdf 2. Thank you for stating the following in the Competing Interests section: "RW, JD, PB, FR and EK were supported by the NIHR Applied Research Collaboration Greater Manchester (NIHR200174) and RW and JD by the NIHR Manchester Biomedical Research Centre (NIHR203308)." Please confirm that this does not alter your adherence to all PLOS ONE policies on sharing data and materials, by including the following statement: ""This does not alter our adherence to  PLOS ONE policies on sharing data and materials.” (as detailed online in our guide for authors http://journals.plos.org/plosone/s/competing-interests).  If there are restrictions on sharing of data and/or materials, please state these. Please note that we cannot proceed with consideration of your article until this information has been declared.  Please include your updated Competing Interests statement in your cover letter; we will change the online submission form on your behalf. 3. Please include your full ethics statement in the ‘Methods’ section of your manuscript file. In your statement, please include the full name of the IRB or ethics committee who approved or waived your study, as well as whether or not you obtained informed written or verbal consent. If consent was waived for your study, please include this information in your statement as well. 4. Please ensure that you refer to Figure 2 in your text as, if accepted, production will need this reference to link the reader to the figure. 5. We note you have included a table to which you do not refer in the text of your manuscript. Please ensure that you refer to Table 3 in your text; if accepted, production will need this reference to link the reader to the Table. 6. Please include captions for your Supporting Information files at the end of your manuscript, and update any in-text citations to match accordingly. Please see our Supporting Information guidelines for more information: http://journals.plos.org/plosone/s/supporting-information.

Reviewers' comments:

Reviewer's Responses to Questions

**Comments to the Author**

1. Is the manuscript technically sound, and do the data support the conclusions?

Reviewer #1: Yes

Reviewer #2: Partly

2. Has the statistical analysis been performed appropriately and rigorously? 

Reviewer #1: Yes

Reviewer #2: I Don't Know

3. Have the authors made all data underlying the findings in their manuscript fully available?

Reviewer #1: Yes

Reviewer #2: No

4. Is the manuscript presented in an intelligible fashion and written in standard English?

Reviewer #1: Yes

Reviewer #2: Yes

5. Review Comments to the Author

Reviewer #1: Excellent manuscript. Congratulations on a hard work well presented. One detail that needs clarification: Was the same device used at home for all patients? Same model and technology for pulse oximetry?

Reviewer #2: The authors present the results of an observational study assessing the effects of home pulse oximeter monitoring among COVID-19 patients.

Overall, the study is interesting. However several limitations should be addressed:

- Although the authors state that the exposure is the pulse oximeter, the patients who accepted to receive a pulse oximeter, actually seem to participate in a program that facilitated the prompt access to the health care system. That is, a more complex intervention, where pulse oximeters are a component.

- The demographic and clinical characteristics used to match cases and controls, probably not controlled all potential confounders. No information about body weight is available.

- It would be helpful to show information about treatments prescribed (both chronic and for the COVID-19 event), at the time considered as baseline.

- It would be informative to show the adjusted survival curves for binary outcomes.

- The limitations of the study, including those of the methods used for causal inferences, should be discussed in a more detailed way.

6. PLOS authors have the option to publish the peer review history of their article (what does this mean?). If published, this will include your full peer review and any attached files.

Reviewer #1: **Yes: **Mahmoud Elfiky

Reviewer #2: No

---

## [Decision Letter · Decision Letter 1]

8 Sep 2024

Patient use of pulse oximetry to support management of COVID-19 in Greater Manchester: a non-randomised evaluation using a target trial approach

PONE-D-24-30539R1

Dear Dr. Kontopantelis,

We’re pleased to inform you that your manuscript has been judged scientifically suitable for publication and will be formally accepted for publication once it meets all outstanding technical requirements.

Kind regards,

Felix Bongomin, MB ChB, MSc, MMed, FECMM

Academic Editor

PLOS ONE

Additional Editor Comments (optional):

Reviewers' comments:

Reviewer's Responses to Questions

**Comments to the Author**

1. If the authors have adequately addressed your comments raised in a previous round of review and you feel that this manuscript is now acceptable for publication, you may indicate that here to bypass the “Comments to the Author” section, enter your conflict of interest statement in the “Confidential to Editor” section, and submit your "Accept" recommendation.

Reviewer #2: All comments have been addressed

2. Is the manuscript technically sound, and do the data support the conclusions?

Reviewer #2: Yes

3. Has the statistical analysis been performed appropriately and rigorously? 

Reviewer #2: Yes

4. Have the authors made all data underlying the findings in their manuscript fully available?

Reviewer #2: No

5. Is the manuscript presented in an intelligible fashion and written in standard English?

Reviewer #2: Yes

6. Review Comments to the Author

Reviewer #2: Thank you for letting me review the revised version of the manuscript.

The authors have appropriately addressed all the comments. Congratulations, great work.

7. PLOS authors have the option to publish the peer review history of their article (what does this mean?). If published, this will include your full peer review and any attached files.

Reviewer #2: **Yes: **Javier Mariani

---

## [Editor Report · Acceptance letter]

14 Sep 2024

PONE-D-24-30539R1 

PLOS ONE

Dear Dr. Kontopantelis, 

I'm pleased to inform you that your manuscript has been deemed suitable for publication in PLOS ONE. Congratulations! Your manuscript is now being handed over to our production team.

Kind regards, 

on behalf of

Dr. Felix Bongomin 

Academic Editor

PLOS ONE